# Experimental and Numerical Study of Low-Velocity Impact and Tensile after Impact for CFRP Laminates Single-Lap Joints Adhesively Bonded Structure

**DOI:** 10.3390/ma14041016

**Published:** 2021-02-21

**Authors:** Chunxing Hu, Guibin Huang, Cheng Li

**Affiliations:** School of Mechanical and Power Engineering, Zhengzhou University, Science Road 100, Zhengzhou 450001, China; huchunxing2020@163.com (C.H.); gbhuang96@163.com (G.H.)

**Keywords:** single-lap joints, low-velocity impact, impact behavior, damage modes, residual strength

## Abstract

To investigate the mechanical behavior of the single-lap joints (SLJs) adhesively bonded structure of carbon fiber reinforced polymer (CFRP) laminates under the low-velocity impact (LVI) and tensile-after impact (TAI), tests and simulations were carried out. A finite element model (FEM) was established based on the cohesive zone model (CZM) and Hashin criterion to predict the damage evolution process of adhesive film, intra- and inter-laminar of the SLJs of CFRP laminates, and its effectiveness was verified by experiments. Moreover, three different overlap lengths (20 mm, 30 mm, and 40 mm) and four different impact energies (Intact joint, 10 J, 20 J, and 30 J) are considered in the present study. Finally, the effects of different impact energies and overlap lengths on the residual strength of SLJs after impact were discussed. The results divulged that numerical results of impact and TAI processes of SLJs were in good agreement with experiment results. During the impact process, the damage of the laminates was primarily fiber and matrix tensile damage, whereas the adhesive film was damaged cohesively; the areas of damage increased with the increase of impact energy, and the normal stress of the adhesive film expanded from the edge to the middle region with the increase of impact force. The influence of LVI on SLJs adhesively bonded structures was very significant, and it is not effective to obtain a higher impact resistance by increasing the overlap length. For the tensile process, the failure mode of TAI of the SLJs was interface failure, the surplus strength of the SLJs gradually decreased with the increase of the impact energy because of the smaller overlap length, the overlap length more than 30 mm, and the low energy impact has almost no effect on the residual strength of the SLJs.

## 1. Introduction

Carbon fiber reinforced polymer (CFRP) laminate single-lap joints (SLJs) structure has become the core part of automobile and aircraft structure because of its small number of components, lightweight design, and high connection efficiency [1,2,3,4]. In the process of fabrication, use, and maintenance, CFRP laminate SLJs adhesively bonded structure are inevitably subjected to low-velocity impact (LVI), resulting in various damages, including matrix damage, fiber damage, and delamination damage, known as barely visible impact damage (BVID) [5,6,7], which severely affects the integrity, residual strength, and service life of SLJs. Borba et al. [8] investigated the effect of different impact energies on the residual strength of friction riveted joints based on an experimental approach. They discovered that small energy impacts produced BVID, which had less effect on the joint strength, while high energy impacts significantly reduced the residual strength of the joint. Therefore, it is important to study the low-velocity impact response and damage characteristics of the SLJs and the residual strength of SLJs with LVI damage to design the damage tolerance of CFRP laminate SLJs adhesively bonded structure.

Experimental and numerical investigations of the LVI behavior of SLJs with metals and dissimilar materials have demonstrated that the impact and tensile-after impact (TAI) behavior and residual strength of SLJs depend on the manufacturing process and a suitable parametric design [9,10,11,12,13,14,15,16,17]. Asgharifar et al. [10] used the three-dimensional dynamic finite element method to study the influence of different adhesive layer thickness, impact energy, and strain rate on the joint stress and verified its effectiveness. The dynamic finite element model established by them provides a reference for the research of adhesively bonded structures. Ma et al. [11] evaluated the effects of overlap length and impact loads on the damage modes and failure loads of CFRP/Al SLJs by an experimental approach. They revealed that as the overlap length increased, the ultimate failure load and impact resistance of the SLJs increased, and the joints mainly appeared as an adhesive film or interface failures. Furthermore, Liu et al. [16,17] investigated the impact resistance and TAI behavior of SLJs adhesively bonded structure of dissimilar materials. They discovered that when the impact surface was the aluminum plate, it had an enhanced effect on the residual strength of SLJs, and when the impact surface was CFRP, the single-lap joint had better impact resistance. The tiebreak approach is also often used to predict damage of composite laminates [18,19]. Zhang et al. [20] developed an impact model (uniform model) for composite materials and then used the tiebreak approach to develop a layered model. They found that the uniform model can simulate the main ballistic impact behavior of composite materials, whereas the layered model can simulate more detailed features of composite damage. Moreover, Li et al. [21] established a finite element model (FEM) based on the cohesive zone model (CZM) and verified its effectiveness by experiments. They found that the stress distribution of the adhesive film was V-shaped and increased toward the edges under impact energies, whereas the residual strength of SLJs decreases significantly when the impact energy increases.

In recent years, the LVI and TAI behaviors of composite laminate SLJs adhesively bonded structures have also been investigated based on the finite element (FE) and several researchers’ experimental methods [22,23,24,25]. Vaidya et al. [26] investigated the effect of transverse impact loads on SLJs of CFRP using experimental and numerical methods. The studies showed that the transverse impact loads resulted in significantly higher stress concentrations in the adhesive film peel than the in-plane loads. Sayman et al. [27] studied the failure of composite adhesively bonded joints under the transverse impact. They observed that the residual strength of the bonded joints decreased when the impact energy increased from 5 J to 15 J. However, at 20 J, the residual strength of the joints increased due to the mutual embedding of the adhesive and adherend. Ergun et al. [28] investigated the flexural behavior of SLJs when three overlap lengths were subjected to different impact energies through an experimental method. They found that the overlap length and the impact energy level greatly affected the impact response of composite laminate SLJs. Liu et al. [29] conducted an experimental and simulation investigation of the impact behavior of scarfed bonding structure of CFRP laminates. The damage mechanism of the internal material was revealed under different impact energies. Meanwhile, they also discovered that the higher the impact energy, the lower the residual strength of the joint.

In summary, the static performance of the CFRP laminate SLJs adhesively bonded structure was systematically analyzed by academics, but the dynamic analysis of the LVI and the residual strength of TAI was not comprehensive and systematic. To investigate more systematically the low-velocity impact behavior and residual strength of single-lap glued structures. In the present study, First, the LVI and TAI model of CFRP laminate single-lap adhesively bonded structure are established under different impact energies based on the triangular CZM and Hashin criterion by using ABAQUS software. Furthermore, an experimental method was used to verify the correctness of the FEM with an overlap length of 20 mm. Finally, the effects of different overlap lengths (20 mm, 30 mm, and 40 mm) and different impact energies (intact joint, 10 J, 20 J, and 30 J) on the impact resistance and residual strength of SLJs are discussed.

## 2. Experimental Procedure

### 2.1. Materials and Specimens Preparation

In this study, composite laminates of unidirectional carbon/epoxy prepreg T300/7901 (Weihai Guangwei Composites Company, Weihai, China) with a thickness of 3.6 mm were used as the adherend material. it was manufactured by a high-pressure process according to a specific stacking sequence [45/0/-45/90]3s. The adhesive of medium temperature curing (120 °C) toughened epoxy resin film LJM-170 was produced by China Weihai Guangwei Composites Company. It has many characteristics, such as a flexible curing cycle, easy handling, and high viscosity. It also has favorable peel and shear resistance over a wide temperature range. The mechanical properties of T300/7901 [30,31] carbon fiber/epoxy resin composite laminate and adhesive film LJM-170 [32] are listed in Table 1, where subscripts 1, 2, and 3 denote longitudinal, in-plane transverse, and through-thickness directions, respectively; c and t denote tensile and compression, respectively.

The three-dimensional geometry of the SLJs structure of the CFRP laminate is shown in Figure 1a. The total length of the structure is 200 mm, the width is 25 mm, the thickness of the adhesive film is 0.2 mm, the radius and mass of the impactor are 12.5 mm and 2.5 Kg, respectively. Aluminum shims of 50 × 25 × 3.6 mm^3^ are bonded to the ends of this structure to reduce the effect of deflection on the specimen during impact and tension. Due to the complexity of the specimen fabrication process and the need to consume a lot of time, we only performed the impact and TAI tests at different impact energies (intact joint, 10 J, 20 J, and 30 J) for the overlap length of 20 mm specimens, and each different impact energy was repeated five times.

The fabrication of SLJs structural specimens of CFRP laminates consists of the following four main processes. First, all the composite adherend materials and aluminum shims were cut to the final size from the bulk plate using the ultra-high pressure water jet cutting machine HSQ1020S according to Figure 1a. Next, to increase the surface roughness of the laminate and improve the adhesion contact area [33], the surface of the area to be connected by the laminate was sanded with sandpaper. The direction of sandpaper sanding significantly affects the bonding strength of the SLJs, and the bonding effect is most significant when the sanding direction is perpendicular to each other and the direction of force [34]. After sanding, the sanded surfaces were cleaned and dried with acetone. Furthermore, to ensure uniform adhesion of the adhesive layer, a steel wire of the same diameter as the thickness of the adhesive layer was inserted into the adhesion area and the parts were assembled and left at room temperature for more than 6 h to ensure sufficient adhesion in the bonding area. Finally, the CFRP laminate SLJs structures were cured in a vacuum oven, maintained the curing temperature at 120 °C for 90 min, and the specimen was cooled at the end, as shown in Figure 1b.

### 2.2. Low-Velocity Impact and Tensile-After Impact Tests

The tests are divided into the LVI and TAI test of this study. A drop-weight impact test was designed according to the ASTM D7136/D7136-07 test standard [35]. Experiments were carried out using XBL-300 (Changchun Kexin Testing Instruments Co, Changchun, China) drop-weight impact test machine at room temperature. Manufacturing initial damage to SLJs by low-velocity impact tests. This low-velocity impact test system includes a hemispherical impactor, a control platform, a data acquisition device, and an anti-secondary impact device. First, the specimen was clamped on a rigid plate (to ensure that the SLJs would avoid complete failure while producing impact damage [9,16], so we redesigned a new test fixture), as illustrated in Figure 2a. Moreover, three different impact energies (10 J, 20 J, and 30 J) were used to evaluate the impact resistance of CFRP laminate SLJs structures, mainly by adjusting the impactor height (0.408 m, 0.816 m, and 1.224 m). In addition, the instantaneous impact force was measured during the test using a kinetic sensor (ICP 208C03), and the force-displacement and energy absorption curves of the impactor were calculated by Equations (1) and (2).
(1)$$d=\int_{0}^{t}\left[v_{0}-\frac{1}{m}\int_{0}^{t}F\left(t\right)dt\right]dt$$
(2)$$E\left(t\right)=E_{k0}-\frac{1}{2}m{\left[v_{0}-\frac{1}{m}\int_{0}^{t}F\left(t\right)dt\right]}^{2}$$

To investigate the effect of different impact energies on the residual strength of the SLJs adhesively bonded structure. The uniaxial tensile test was designed according to the ASTM D1002 test standard [36], Experiments were carried out using WDW-300 (Changchun Kexin Testing Instruments Co, Changchun, China) universal tensile test machine. The relative error of test force is less than 1%. The clamped ends of the specimen were fixed by the knurling of the tensioner jig to prevent slippage. The lower end of the specimens was solidly supported and the upper end was moved with the beam by applying an axial load of 2 mm/min. The data of load and displacement during the tensile process are collected by a computer, as shown in Figure 2b.

## 3. Numerical Methods

As described in Section 2.2, the force-time curves and ultimate failure loads of SLJs are obtained from LVI and TAI tests, respectively. However, in the above test plan, only three impact energies with an overlap length of 20 mm are considered. Therefore, to further understand the effect of overlap lengths (20 mm, 30 mm, and 40 mm) on the impact resistance and residual strength of SLJs adhesively bonded structures of composite laminates under the low-velocity impact, numerical modeling is considered. Furthermore, the details of the material constitutive model used will be described, including the Hashin criterion and the cohesion zone model.

### 3.1. Intralaminar Damage Criterion

The Hashin failure criterion can accurately predict various failure modes and is simple and effective. Meanwhile, in-plane damage failure is mainly a two-dimensional intralaminar problem. Therefore, in this study, the Hashin criterion [37] is used to predict the intralaminar failure behavior of CFRP laminates, given in Equations (3)–(6).

Fiber tensile damage ($\sigma_{11}\geq 0$),
(3)$${\left(\frac{\sigma_{11}}{X_{t}}\right)}^{2}+{\left(\frac{\sigma_{12}}{S_{12}}\right)}^{2}\geq 1$$

Fiber compressive failure ($\sigma_{11}<0$),
(4)$${\left(\frac{\sigma_{11}}{X_{c}}\right)}^{2}\geq 1$$

Matrix tensile failure ($\sigma_{22}\geq 0$),
(5)$$\frac{{\left(\sigma_{22}\right)}^{2}}{Y_{t}^{2}}+{\left(\frac{\sigma_{12}}{S_{12}}\right)}^{2}\geq 1$$

Matrix compressive failure ($\sigma_{22}<0$),
(6)$$\frac{\sigma_{22}}{Y_{c}}\left[{\left(\frac{Y_{c}}{2S_{23}}\right)}^{2}-1\right]+{\left(\frac{\sigma_{12}}{S_{12}}\right)}^{2}+{\left(\frac{\sigma_{22}}{2S_{23}}\right)}^{2}\geq 1$$
where $\sigma_{11}$ and $\sigma_{22}$ are the normal stresses in the x and y directions, respectively, and $\sigma_{12}$ is the shear stress. Moreover, *X_t,c_*, *Y_t,c_*, and *S* are given in Table 1.

When an element satisfies the failure criterion in the model, it is considered to be damaged and the material property parameters of this element will be discounted and the stress-strain distribution will be changed. Therefore, a damage evolution law based on continuum damage mechanics (CDM) is used in Abaqus/Explicit to determine the degradation of material properties for each damage mode. When damage occurs in a CFRP laminate, the constitutive relation is defined as [38,39]
(7)$$\sigma =C_{d}\varepsilon =\frac{1}{D_{f}}\left[\begin{array}{ccc} \left(1-d_{f}\right)E_{1} & \left(1-d_{f}\right)\left(1-d_{m}\right)\upsilon_{21}E_{1} & 0\\ \left(1-d_{f}\right)\left(1-d_{m}\right)\upsilon_{12}E_{2} & \left(1-d_{m}\right)E_{2} & 0\\ 0 & 0 & \left(1-d_{s}\right)G_{12}D_{f} \end{array}\right]\left[\begin{array}{c} \varepsilon_{11}\\ \varepsilon_{22}\\ \varepsilon_{33} \end{array}\right]$$
where $D_{f}=1-\left(1-d_{f}\right)\left(1-d_{m}\right)\upsilon_{12}\upsilon_{21}$, $G_{12}$ represents the shear modulus, $d_{f}$, $d_{m}$, and $d_{s}$ are the damage variables fiber, matrix, and shear damage, respectively, and the expressions are as follows [39]:(8)$$d_{f}=\{\begin{array}{c} d_{f}^{t}\;\sigma_{11}\geq 0\\ d_{f}^{c}\;\sigma_{11}<0 \end{array}$$
(9)$$d_{m}=\{\begin{array}{c} d_{m}^{t}\;\sigma_{22}\geq 0\\ d_{m}^{c}\;\sigma_{22}<0 \end{array}$$
(10)$$d_{s}=1-\left(1-d_{f}^{t}\right)\left(1-d_{f}^{c}\right)\left(1-d_{m}^{t}\right)\left(1-d_{m}^{c}\right)$$

The damage variables $d_{f}^{t}$, $d_{f}^{c}$, $d_{m}^{t}$, and $d_{m}^{c}$ correspond to the four damage modes of the Hashin failure criterion, which are important parameters to describe the damage status.

During material damage evolution, the following damage factor *d* is introduced to describe the damage state of the material at a moment:(11)$$d=\frac{\delta_{eq}^{f}\left(\delta_{eq}-\delta_{eq}^{0}\right)}{\delta_{eq}\left(\delta_{eq}^{f}-\delta_{eq}^{0}\right)}$$
where $\delta_{eq}^{f}$ denotes the final equivalent displacement when the composite is completely damaged in the corresponding mode. $\delta_{eq}^{0}$ and $\delta_{eq}$ are the equivalent displacements for reaching the initial damage and any moment, respectively. $d_{f}$,$\;d_{m}$, and $d_{s}$ can be calculated based on the equivalent displacement corresponding to each damage mode. In this way, the degree of damage to the element in different damage modes can be clearly obtained during the finite element analysis.

### 3.2. Interlaminar and Adhesive Damage Criterion

The primary form of interlaminar damage is delamination in CFRP laminates, i.e., debonding between sub-layers. In this study, the cohesive element in the CZM is used to simulate the interface layer, and the failure and damage process of the interface layer element is used to simulate the initiation and expansion process of the delamination damage. The bilinear cohesion response model is shown in Figure 3.

The damage initiation between the adhesive layer and the CFRP laminate layers is predicted using a quadratic nominal stress criterion [40].
(12)$${\left\{\frac{\langle t_{n}\rangle }{t_{n}^{0}}\right\}}^{2}+{\left\{\frac{t_{s}}{t_{s}^{0}}\right\}}^{2}+{\left\{\frac{t_{t}}{t_{t}^{0}}\right\}}^{2}=1$$
where $t_{n}^{0}$ is the normal strength of the CFRP interlaminar and adhesive in the third direction; B and C are the shear strengths. When the cohesive element is not damaged, its linear elastic constitutive relationship is as follows:(13)$$t=diag\left(K_{nn},K_{ss},K_{tt}\right)\delta$$
where $K_{nn}$, $K_{ss}$ and $K_{tt}$ are the normal tensile, in-plane, and out-plane shear stiffnesses, respectively. δ indicates separation displacement. When Equation (12) is satisfied, the cohesive element starts to fail. the constitutive relation of the damaged cohesive element is
(14)$$t=\left(1-D\right)diag\left(K_{nn},K_{ss},K_{tt}\right)\delta$$
where D is the damage factor, and its range is [0,1]. When D = 0, the material has not been damaged; when D = 1, the stiffness of the material has degraded to zero. It is used to represent the energy released during the degradation of stiffness and final failure of the material and reaches a critical value when the material fails.

The damage and evolution processes of cohesive elements were predicted using the B–K fracture criterion based on the hybrid cracking model proposed by Benzeggagh and Kenane [38].
(15)$$G^{C}=G_{n}^{C}+\left(G_{s}^{C}-G_{n}^{C}\right){\left(\frac{G_{s}+G_{t}}{G_{n}+G_{s}}\right)}^{\eta }$$
where $G_{n}^{C}$, $G_{s}^{C}$, and $G_{t}^{C}$ are the normal, in-plane, and out-plane shear critical fracture energy, respectively. η is the relevant parameter of the composite material, for carbon fiber epoxy composites; η = 1~2, and in this study, η = 1.45 [41,42].

### 3.3. Finite Element Modeling

To obtain a comprehensive understanding of the impact resistance and residual strength of SLJs adhesively bonded structures of CFRP laminate at LVI and TAI, a 3D FEM is established using ABAQUS/Explicit software based on the CFRP laminate intralaminar and interlaminar and adhesive failure models, as shown in Figure 4. The layers are each 0.15 mm thick 24-layer composite layup. The laminate was formed by discrete eight-node continuous shell element SC8R, and the interlaminar and adhesive were formed by discrete COH3D8 element. In this study, the CFRP laminate was meshed layer by layer, the mesh was divided into structural meshes and the mesh at the joints is refinement. The grid unit size at the joints: 1 mm × 1 mm × 0.15 mm for length × width × thickness, and 2 mm for the rest of the dimensions. Cohesive elements of zero-thickness were inserted between the laminate layers to predict delamination damage.

Firstly, the numerical simulation of LVI was conducted using the FEM shown in Figure 4. During the impact process, a general contact was used between the impactor and the SLJs, and friction was generated throughout the contact. The friction coefficient was set to 0.1 for all contact points, the tangential behavior was set to penalty function contact, and the normal behavior was hard on contact. The impactor was assumed a rigid body in the impact process. To achieve different impact energies, the prescript mass and downward initial velocity for the impactor were set. Moreover, the ends and lower surfaces of the single-lap joint adhesively bonded structures were completely fixed.

Subsequently, for the second analysis step, the SLJs adhesively bonded structure of CFRP laminate with LVI damage was calculated by the data transfer module. During the axial stretching process (the boundary conditions are shown in Figure 4), the left end of the structure was entirely fixed, the reference point RP-2 was established on the right side, and all nodes on the end face were coupled to the reference point. A displacement load along the X direction was applied at the reference point.

## 4. Results and Discussion

In this study, to better investigate the impact and TAI properties of CFRP laminate SLJs adhesively bonded structures, experimental and numerical tests of three different overlap length joints and impact energy levels were performed using a drop-weight test machine (XBL-300), a universal tensile machine (WDW-300, and ABAQUS simulation software. The impact responses, such as contact loading, deflection, and energy absorption, in addition to failure mechanisms of SLJs, were analyzed and discussed.

### 4.1. Impact Response Analysis of SLJs

#### 4.1.1. Effect of Impact Energy on the Impact Response of SLJs

In this study, three different impact energies were used to verify the accuracy of the FEM by conducting low-velocity impact tests on SLJs adhesively bonded structure of CFRP laminate with *L* = 20 mm. Figure 5a–c shows the force–time curves recorded by the sensor at impact energies of 10 J, 20 J, and 30 J. As can be seen from the figure, the impact test is divided into three main phases. In the first phase, when the impactor contacted the SLJs, no damage occurred to the SLJs. In the second phase, the CFRP laminate began to fail, and the impact force begins to oscillate. When the impact force peaked, the oscillation was the most intensive, which indicates that the damage was severe at this time. The final phase is the rebound process of the impactor, where the impactor rebounded under the laminate action, and the impact force gradually decreased to zero.

In order to compare with the simulation results more clearly, Figure 5a–c shows the test force–time curve with the peak force closest to the average of the five sets of test results. It is found that the peak force increases gradually with increasing impact energy. The peak forces of the experimental results are 8.78 kN, 13.70 kN, and 17.29 kN under the three energy impacts, the peak forces of the simulation results are 9.3 kN, 14.01 kN, and 18.64 kN, and the errors of the experimental and simulation results are 5.92%, 2.26%, and 7.8%, respectively, which confirms the validity of the finite element model. When the impact energy reaches 30 J, the trend of the test and simulation results is not consistent. The reason may be that the damage internal the joint is more serious and the curve oscillation is more frequent when the joint is impacted by higher energy. Moreover, the force–time curves of the simulation results are almost all above the experimental results. The cause of the error may be the wear between the drop-weight and the guide rail during the falling process and the non-uniform distribution of the adhesive layer in the bonding area during the production process of the SLJs, in addition to bubbles, micro-cracks, and other internal original defects.

The load–displacement curve is the response to the impact load of the SLJs adhesively bonded structure of CFRP laminate. This response obtained important information about the LVI behavior of the SLJs, as calculated by Equation (1). For comparison, the curves of the impact force–displacement of the SLJs adhesively bonded structure of CFRP laminate under different impact energies are illustrated in Figure 6a–c. The figure shows that the load and displacement of each curve are divided into two phases—rising and falling. The rising phase is caused by the resistance of the SLJs to the impact load, and the falling phase indicated the impactor rebounded from the surface of the SLJs. As the impact energy increases, the contact load and the displacement of the impactor gradually increase. The displacements of 20 J and 30 J impact energies compared to the 10 J increased by 25.63% and 55%, respectively. Moreover, these curves have closed shape characteristics, which provide a theoretical basis for the energy absorption in SLJs [43,44].

The absorbed energy–time curve of the SLJs adhesively bonded structure of CFRP laminate is calculated from Equation (2), as present in Figure 7. As shown in the figure, the energy absorption–time curve rises with time at the beginning of the process, reaches a peak, and then decreases and maintains steadily. The peak value indicates the different impact energies, and the final constant value represents the final energy absorbed by the specimen. When the impact energy increases, the energy absorbed by the specimen also gradually increases. This law can be explained as follows: when the impact energy of the impactor is small, most of the kinetic energy of the impactor is converted into the elastic energy of the laminate after impacting the SLJs, and then the impactor rebounds and is converted into the kinetic energy of the impactor, so the joint absorbs less energy and suffers less damage. When the impact energy is further increased, the joint damage becomes more extensive, and the adhesive layer will damage to different degrees. Only a small part of the elastic energy is converted into the kinetic energy of the impactor, and the joint and the adhesive layer absorb more energy. The absorbed energy values of the test results under the impact of the three energies (10 J, 20 J, and 30 J) are 7.43 J, 13.76 J, and 21.84 J, respectively, and the absorbed energy of the simulation results are 7.05 J, 14.85 J, and 23.61 J, respectively. The errors between the experimental and numerical results were less than 8.21%.

#### 4.1.2. Effect of Overlap Length on the Impact Response of SLJs

As described in Section 4.1.1., the impact resistance of SLJs adhesively bonded structure of CFRP laminate under LVI varies with the impact energy. Simulation results of different overlap lengths and impact energies on the impact resistance of SLJs adhesively bonded structures are given in Figure 8. From Figure 8a,b, it can be seen that the peak force and energy absorption of the SLJs increase with the impact energy for the same overlap length. As shown in Figure 8a, when the impact energy is 10 J, the peak force of the SLJs adhesively bonded structure increases with the increase of the overlap length, and when the overlap length is increased from 30 mm to 40 mm, the peak force is slightly decreased. When the impact energy is greater than 10 J, the peak force is almost unchanged by increasing the overlap length. As shown in Figure 8b, when the impact energy is 10 J, the energy absorption of the SLJs is almost constant. In contrast, the impact energy is greater than 10 J, the absorbed energy decreases slightly with the increase of the overlap length. The main reason for the above phenomenon is that different overlap lengths cause almost the same damage area to the SLJs at the identical impact energy. In summary, increasing the overlap length cannot effectively increase the impact resistance of the CFRP laminate SLJs [11].

### 4.2. Impact Damage Results and Analysis of SLJs

The impact damage parameters include intra- and inter-laminar damage of CFRP adherend parts and damage of the adhesive film. Among them, the intralaminar damage of CFRP bonded parts includes fiber and matrix damage, the interlaminar damage of CFRP bonded parts are in the form of delamination, and the damage of adhesive layer is in the form of cohesive damage. This section mainly discussed and analyzed the damage behavior of the SLJs with overlap length *L* = 20 mm under different impact energies on the surface and internal of the joints.

#### 4.2.1. Surface Damage Results and Analysis of SLJs

To investigate the damage behavior of the surface of SLJs under different impact energies, Figure 9 indicates the surface damage diagram of the SLJs adhesively bonded structure with overlap length *L* = 20 mm under different impact energies in the experiment. When subjected to impact loading, the pressing of the impactor in a short period will cause a sizeable concentrated force on the CFRP laminate to cause different degrees of damage to the joints’ surface. Due to the minor degree of damage, the damage results are illuminated by visible light. For the impact surface, the joints subjected to impact the action has about 0.05–0.15 mm pits in the central region, and the depth of the pits gradually increases with the increase of impact energy. When the impact energy is 10 J, the damage of the joint surface is barely visible damage (BVID); in comparison, when the impact energy is greater than 10 J, the depth and the diameter of the joint pits gradually increase. The damage of the surface of the joint is mainly fiber and matrix damage. The joint’s bottom was fixed entirely, resulting in almost identical damage to the joint’s backside.

When the surface of the SLJs is subjected to an impact force, tensile stress will be generated in the impact area, resulting in fiber tensile damage. Simulation results of fiber tensile damage of CFRP laminate SLJs at different impact energies are illustrated in Figure 10. The Hashin failure criterion shows that when Equation (3) is higher than one, the fiber fails in tension, and when Equation (3) is equal to zero, the fiber does not show damage. It can be seen from the figure that the location of fiber tensile damage is concentrated around the impact center, and the damage starts along the 45° direction, mainly because the first layer of the laminate is 45°. The fiber stretching appeared in different degrees of damage, and the area gradually increased with the increase of impact energy, but complete damage did not occur, which agrees with the test results in Figure 9.

The matrix of the CFRP adherend part is epoxy resin, and the matrix will crack when subjected to impact load, that is, the matrix is tensile damaged. Simulation results of matrix tensile damage of CFRP laminate SLJs at different impact energies are given in Figure 11. Hashin failure criterion shows that when Equation (5) is higher than one, the matrix fails in tension, and when Equation (5) is equal to zero, the matrix does not show damage. The figure shows that a wide range of tensile damage has occurred to the substrate, and the damage location is also concentrated around the impact center region and the damage initiation and evolution along the −45° direction, which is because the tensile damage of the third layer is higher than that of the first layer [45], while the second layer of 0° ply has strong tensile properties [46], and the degree of damage is small. As the impact energy increases, the area of tensile damage of the matrix became increasingly larger. Because the matrix tensile strength is far lower than the fiber tensile strength, the matrix tensile damage area is much larger than the fiber tensile damaged area. There is no complete damage, which is consistent with the test results in Figure 9.

#### 4.2.2. Internal Damage Results and Analysis of SLJs

To investigate the damage behavior of the internal of SLJs under different impact energies, the impacted SLJs adhesively bonded structure was cut along the transverse direction from the center of the impact area using the ultra-high pressure water jet cutting machine HSQ1020S, and the cut specimen was observed by a high-magnification ultra-depth-of-field microscopy system model VHX-6000 (Vanke Instrument Co, Jiangshu, China). Figure 12 shows the microscopic damage diagram of the SLJs adhesively bonded structure with an overlap length *L* = 20 mm under different impact energies. Figure 12a shows the specimen without impact load, there is no internal damage, and some initial defects internal to the adhesive boned joint can be observed. Figure 12b–d shows that when the impact energy is small, the internal of the laminate is mainly fiber damage and delamination damage, while the adhesive layer is almost not damaged. When the impact energy gradually increases, the internal of the laminate mainly matrix damage, fiber damage, and delamination damage, and the degree of damage is increasing, and the damage of the adhesive layer gradually starts from the edge and expands to the interior.

Typical axial tensile experiments were performed for specimens of SLJs adhesively bonded structures under different energy impacts. The tensile failure mode of the test piece of CFRP laminate SLJs with an overlap length *L* = 20 mm is shown in Figure 13. The failure modes of SLJs are mainly cohesive damage of the adhesive layer and interface failure. The red circled area in the figure is the impact damage area of the impactor. Figure 13 shows that when the impact energy is 10 J, the adhesive layer is hardly damaged. With the increase of impact energy, the damage of the adhesive layer in the center area of the impact becomes more and more serious. Moreover, the residual tensile failure mode of CFRP laminate SLJs remained consistent with intact joints, which is the primarily cohesive failure of the adhesive layer. It showed that regardless of whether it was an intact joint or a joint containing LVI damage, their weak link under tensile loading was the adhesive film.

In ABAQUS software, the SDEG index is used to view the damage evolution status of the interlaminar and adhesive layer. When the value of SDEG is zero, there is no damage to the element; when SDEG is greater than 0, the element begins to occur damage; and when SDEG = 1, the element fails completely. After the CFRP laminate SLJs were subjected to LVI, the adhesive layer in the lap region of the adherend will be damaged; the damage of the adhesive layer has a non-negligible effect on the residual strength of the SLJs. Simulation results of damage to the adhesive layer of the CFRP laminate SLJs at different impact energies are shown in Figure 14. Cohesive damage occurred in the adhesive layer at all three impact energies, and the damaged area increased with the increase of impact energy. When the impact energy is 10 J, the SDEG index of the adhesive layer is 0, that is, the adhesive layer has not yet failed elements; when the impact energy is greater than 10 J, the SDEG index of the adhesive layer in the center area of the impact gradually increases, and the greater the impact energy is, the larger the damaged area. The damage initiation and evolution of the adhesive layer is also along the direction of 45°, mainly because the layup of the adherend parts adjacent to the adhesive layer are all 45°.

Delamination damage is also an important indicator for predicting the impact resistance of SLJs adhesively bonded structures. Figure 15 illustrates simulation results of delamination damage of CFRP laminate SLJs under different impact energies. It can be seen that the delamination damage of the CFRP laminate SLJs under LVI is more serious, and the delamination damage position appeared in the impact center and extends to the surroundings. The delamination damage distribution and areas of each interlaminar element are different, and all damage elements need to be superimposed together to obtain the overall delamination area. With the increase of impact energy, the damaged area also increased, and the degree of delamination damage also gradually increased, which is consistent with the conclusions of the literature [29,47].

### 4.3. Stress Distribution of Adhesive Layer of SLJs under Impact Load

Figure 16 presents the normal stress distribution of the SLJs adhesively bonded structure at different loading points of the adhesive layer under 30 J impact in simulation. Figure 16a–c shows the normal stress distribution at the three loading points of 0.2*F*, 0.6*F*, and *F*, respectively. The figure shows that the normal stresses at different loading points are approximately V-shaped and symmetrically distributed along the overlap length and width, respectively. Stress concentrations were found on the edges and gradually expanded to the middle region with increasing loading points [10,21], which is consistent with the phenomenon in Figure 12.

### 4.4. Residual Strength Analysis of SLJs

Residual strength is used to describe the impact resistance of CFRP laminate SLJs adhesively bonded structures. The influence of different impact energies and overlap lengths on the residual strength of SLJs is discussed and analyzed in this section.

#### 4.4.1. Influence of Impact Energy on Residual Strength of SLJs

A specimen with an overlap length *L* = 20 mm containing LVI damage was subjected to axial tensile testing to verify the correctness of the second step of the FEM. The ultimate failure loads of the SLJs adhesively bonded structure under different impact energies are shown in Figure 17 for both test and simulation. The error of both experimental and simulation results is less than 8.25% observed from the figure, as it verified the correctness of the second step of FEM. When the impact energy is 10 J and 20 J, the residual strength of the joints decreased by 10.11% and 13.65%, respectively, compared with the lossless SLJs; when the impact energy is 30 J, the residual strength of the joints decreased by 23%. Figure 14 observes that the adhesive layer is the weak part of SLJs—in low impact energy, the adhesive layer has almost no damage, and the damage is mainly for the laminate. In contrast, with more considerable impact energy, the adhesive layer’s central area is almost completely damaged, so the residual strength of the joint is significantly reduced.

Figure 18 shows the load–displacement curves of CFRP laminate SLJs with an overlap length of 20 mm in axial tension at different impact energies in simulation. As the impact energy increased, both the failure load and displacement of the joint decreased gradually. The failure displacements are reduced by 0.83%, 5.49%, and 15.11%, respectively, compared to the lossless SLJs. The slope of the curve in the figure represents the tensile stiffness of SLJs. It is obvious that the tensile stiffness decreases with increasing impact energy. The load–displacement curves for 10 J, 20 J, and 30 J also show fluctuations due to the damage internal the CFRP laminate after LVI and the complete degradation of the stiffness of some elements, resulting in a change in the continuity of FEM.

#### 4.4.2. Influence of Overlap Length on Residual Strength of SLJs

The residual strengths of CFRP laminate SLJs adhesively bonded structure with different overlap lengths after impact with different impact energies are shown in Figure 19. The strength of SLJs decreases for three overlap length joints after LVI compared to the lossless joints. The residual strength of SLJs decreases significantly for overlap lengths of 20 mm and 30 mm under different impact energies. At an overlap length of 40 mm, the residual strength of SLJs remains stable under small impact energy. Because the laminate and the adhesive layer showed a mechanical interlocking phenomenon under low impact energy, which did not lead to a significant decrease in joint strength [48]. Therefore, CFRP laminate SLJs with smaller overlap lengths are more sensitive to LVI damage. Beyond 30 mm, the residual strength of SLJs remains practically unchanged compared to the strength of lossless joints under smaller impact energies. Merely increasing the overlap length cannot significantly increase the strength of SLJs.

## 5. Conclusions

In this study, the low-velocity impact behavior and the residual strength after LVI of CFRP laminate SLJs adhesively bonded structures with different overlap lengths (20 mm, 30 mm, and 40 mm) at different impact energies (10 J, 20 J, and 30 J) were investigated by experimental and numerical methods, and the main conclusions are as follows:A FEM was established based on the Hashin failure criterion and CZM, and the experimental analysis of different impact loads was carried out on the joint with an overlap length of 20 mm. The errors of peak force and energy absorption for both experimental and simulation results are less than 10%, and the validity of the finite element method was verified;When the overlap length is fixed, the impact behavior of SLJs, such as peak force, displacement, and energy absorption, increases with the increase in impact energy. When the impact energy is certain, increasing the overlap length cannot effectively improve the impact performance of SLJs;The failure models of SLJs subjected to low-velocity impact are mainly include matrix cracking, fiber damage, and delamination damage. The delamination damage is the most obvious, and the damage areas gradually increase with the increase of impact energy. The stress distribution in the adhesive layer is mainly V-shaped and symmetrical. The stress concentration phenomenon expands from the edge to the middle with the increase of the loading point;For the residual strength of the single-lap adhesively bonded structure at low-velocity impact, when the overlap length is 20 mm, the residual strength of SLJs gradually decreases with impact energy. When the overlap length is 40 mm, the residual strength of the CFRP laminate SLJs remains essentially unchanged compared to the lossless joint at smaller impact energy. Merely increasing the overlap length cannot significantly increase the strength of SLJs. Therefore, CFRP laminate SLJs adhesively bonded structures with smaller overlap lengths are more sensitive to low-velocity impact behavior.

## Figures and Tables

**Figure 1 materials-14-01016-f001:**
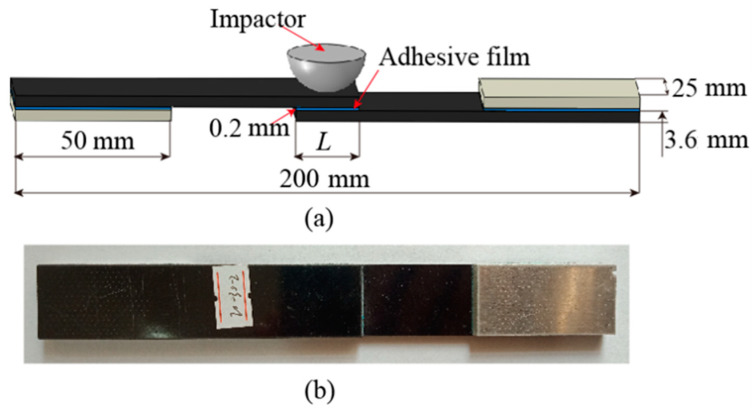
(**a**) Three-dimensional geometry of the single-lap joints (SLJs) (unit: mm) and (**b**) photograph of the specimen.

**Figure 2 materials-14-01016-f002:**
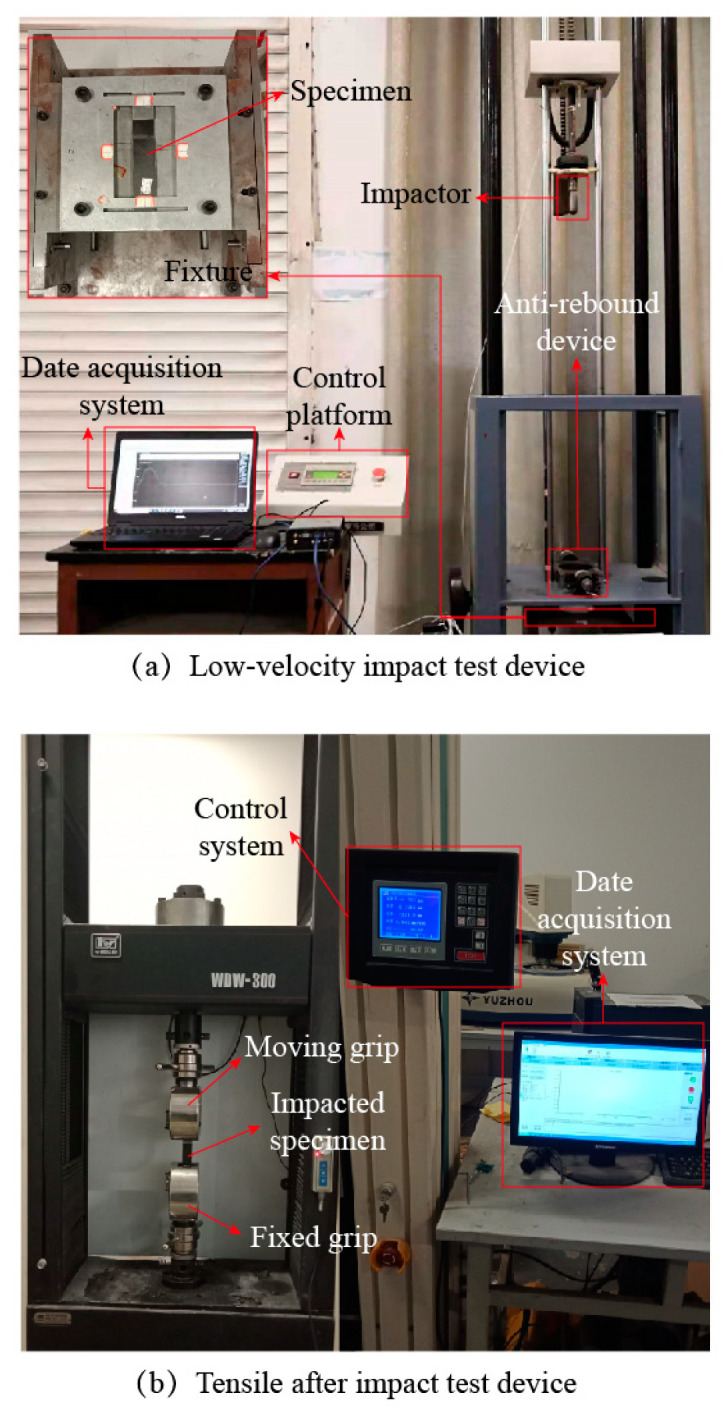
(**a**) Low-velocity impact test device and (**b**) tensile-after impact test devices.

**Figure 3 materials-14-01016-f003:**
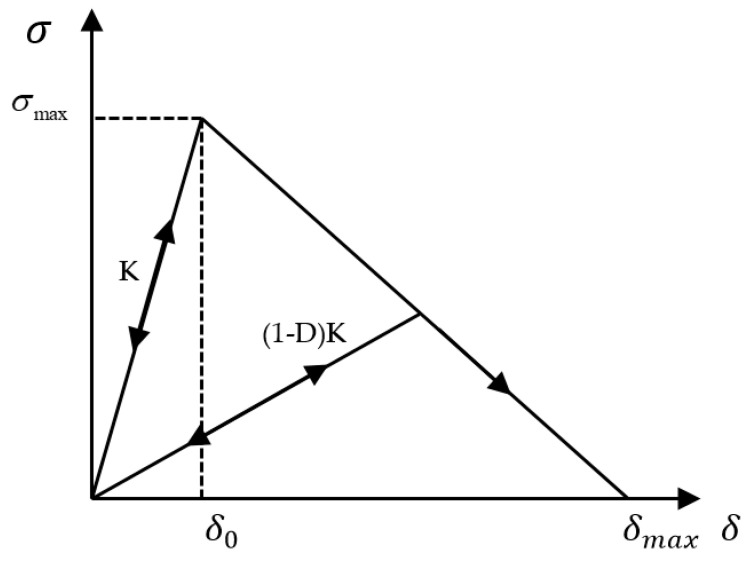
Bilinear constitutive model of the cohesive element.

**Figure 4 materials-14-01016-f004:**
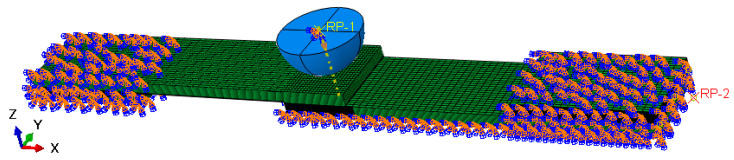
The finite element model (FEM) of LVI and TAI of carbon fiber reinforced polymer (CFRP) laminate SLJs adhesively bonded structures.

**Figure 5 materials-14-01016-f005:**
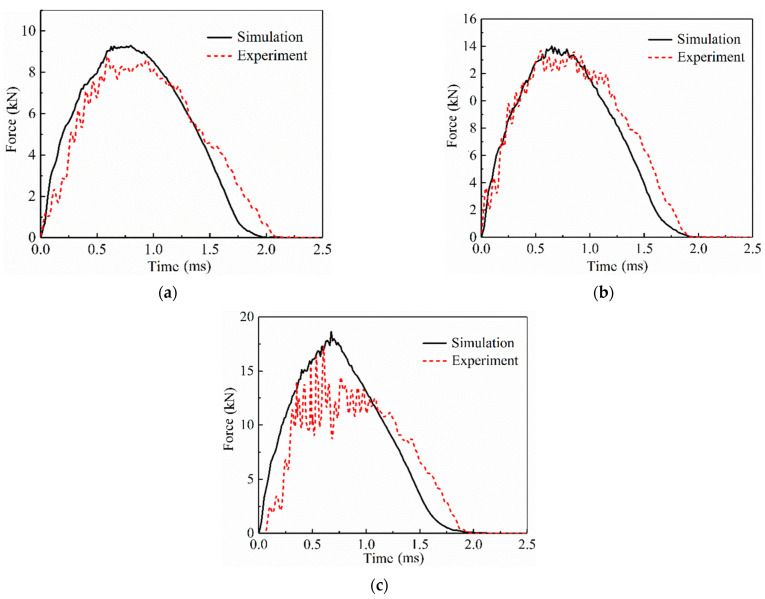
Curves of the impact force-time of the SLJs adhesively bonded structure of CFRP laminate under different impact energies. (**a**) 10 J; (**b**) 20 J; and (**c**) 30 J.

**Figure 6 materials-14-01016-f006:**
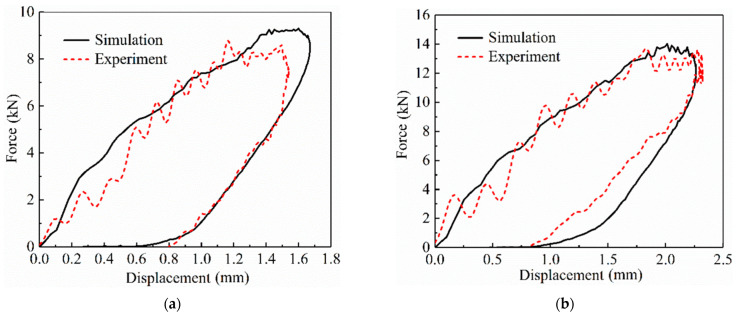
Curves of the impact force-displacement of the SLJs adhesively bonded structure of CFRP laminate under different impact energies. (**a**) 10 J; (**b**) 20 J; and (**c**) 30 J.

**Figure 7 materials-14-01016-f007:**
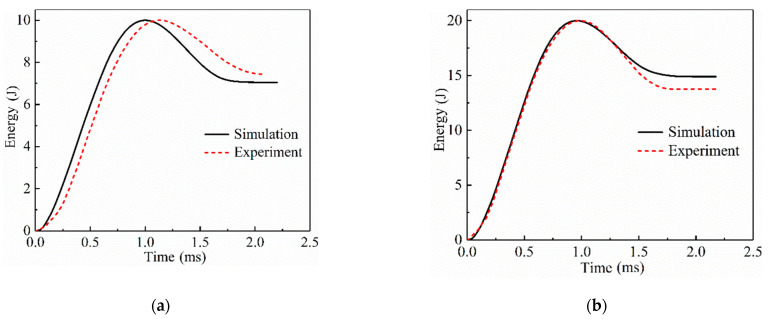
Curves of the energy absorption-time of the SLJs adhesively bonded structure of CFRP laminate under different impact energies. (**a**) 10 J; (**b**) 20 J; (**c**) 30 J.

**Figure 8 materials-14-01016-f008:**
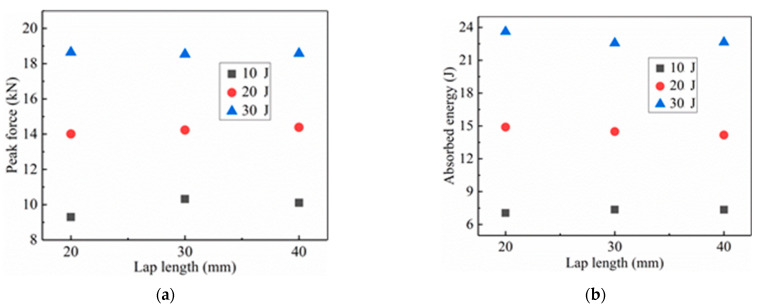
The influence of different overlap lengths and impact energies on the impact resistance of SLJs: (**a**) Peak force at different impact energies and (**b**) Absorbed energy at different impact energies.

**Figure 9 materials-14-01016-f009:**
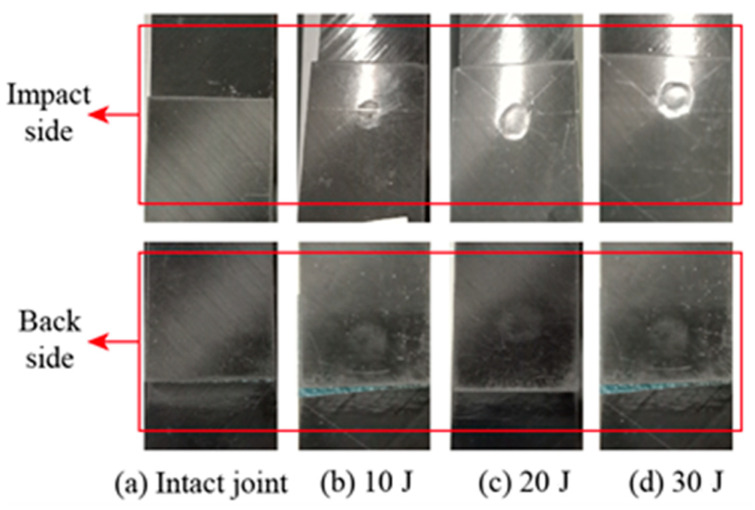
The surface damage of the SLJs under different impact energies: (**a**) Intact joint; (**b**) 10 J; (**c**) 20 J; (**d**) 30 J.

**Figure 10 materials-14-01016-f010:**
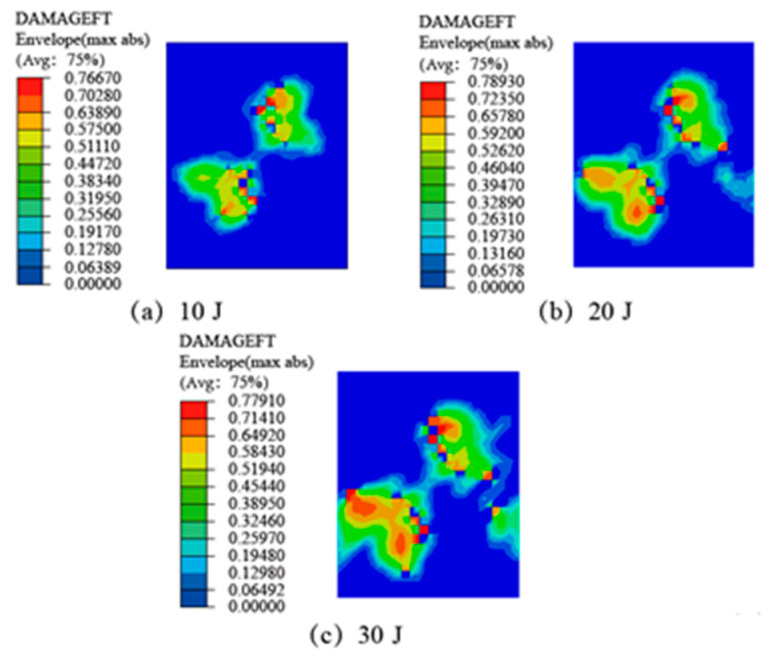
The fiber tensile damage of CFRP laminate SLJs at different impact energies: (**a**) 10 J; (**b**) 20 J; (**c**) 30 J.

**Figure 11 materials-14-01016-f011:**
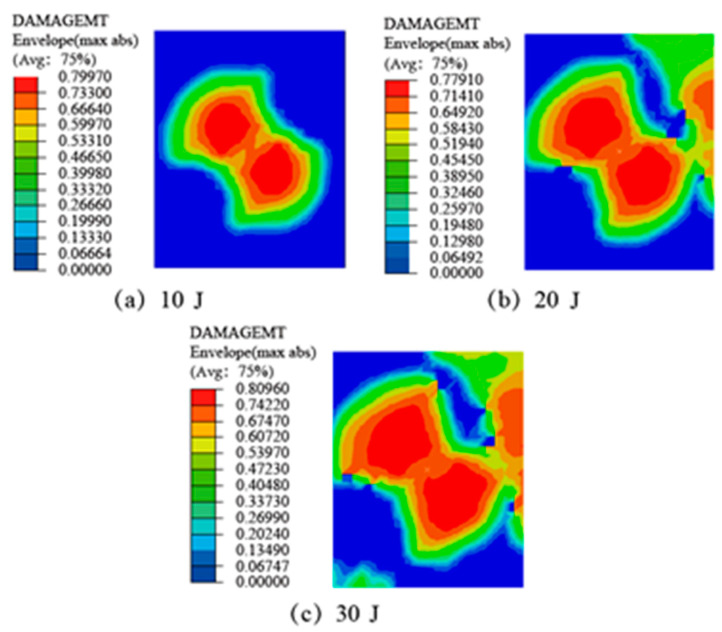
The matrix tensile damage of CFRP laminate SLJs at different impact energies.: (**a**) 10 J; (**b**) 20 J; (**c**) 30 J.

**Figure 12 materials-14-01016-f012:**
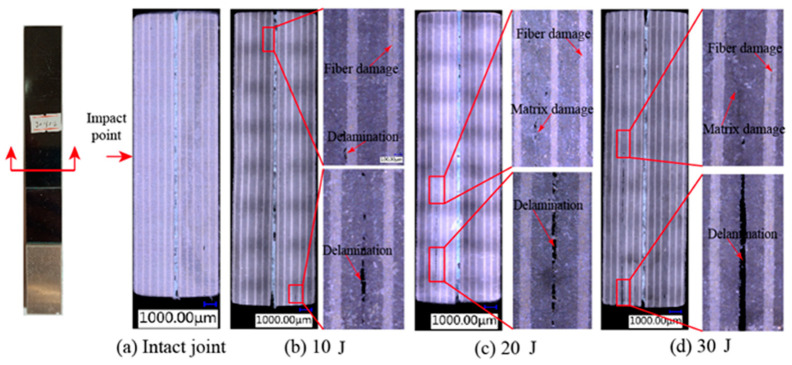
The microscopic damage of CFRP laminate SLJs at different impact energies: (**a**) Intact joint, (**b**) 10 J, (**c**) 20 J, (**d**) 30 J.

**Figure 13 materials-14-01016-f013:**
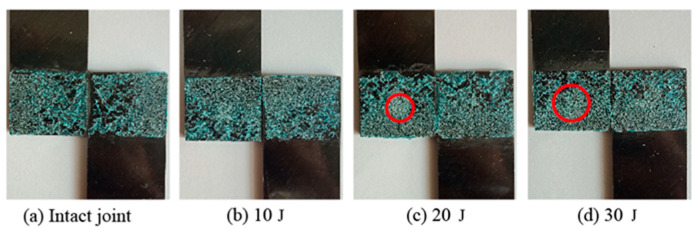
The adhesive film damage of TAI of CFRP laminate SLJs: (**a**) Intact joint, (**b**) 10 J, (**c**) 20 J, (**d**) 30 J.

**Figure 14 materials-14-01016-f014:**
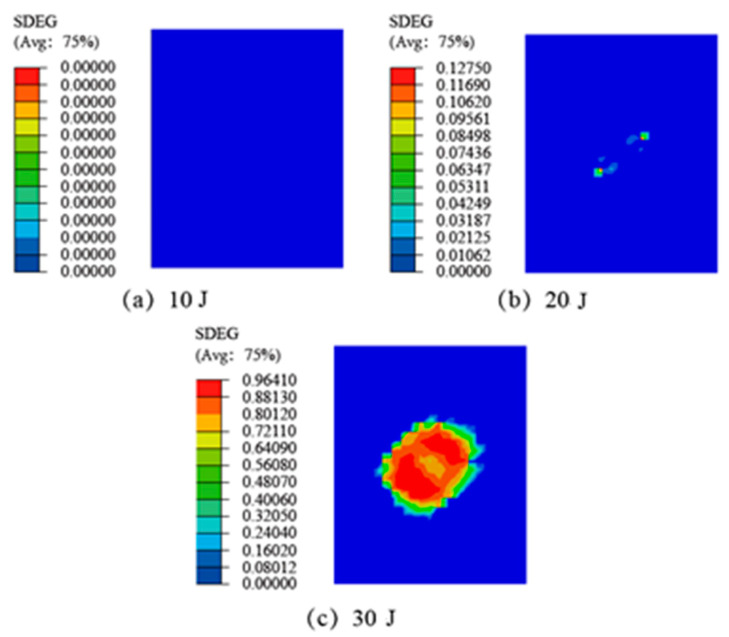
The adhesive layer impacts damage of the CFRP laminate SLJs at different impact energies: (**a**) 10 J, (**b**) 20 J, (**c**) 30 J.

**Figure 15 materials-14-01016-f015:**
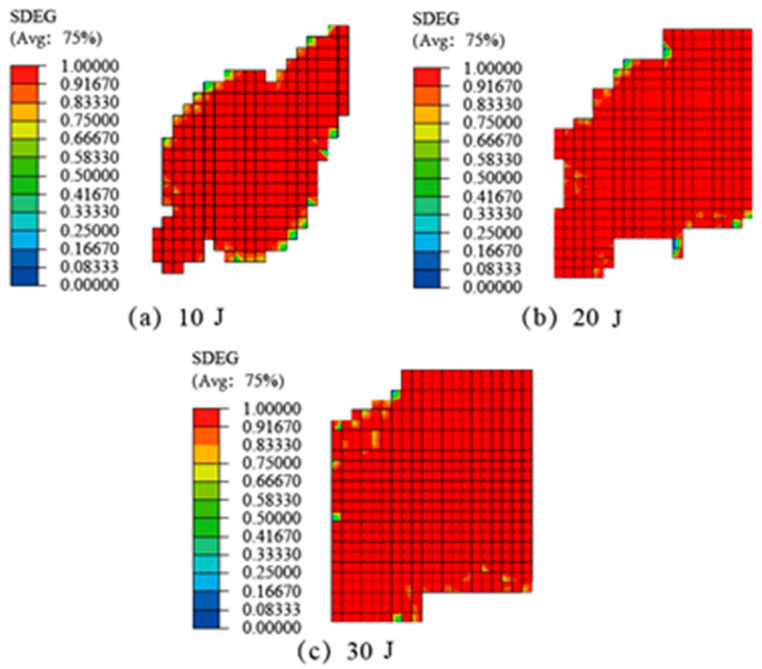
The delamination damage of CFRP laminate SLJs at different impact energies: (**a**) 10 J, (**b**) 20 J, (**c**) 30 J.

**Figure 16 materials-14-01016-f016:**
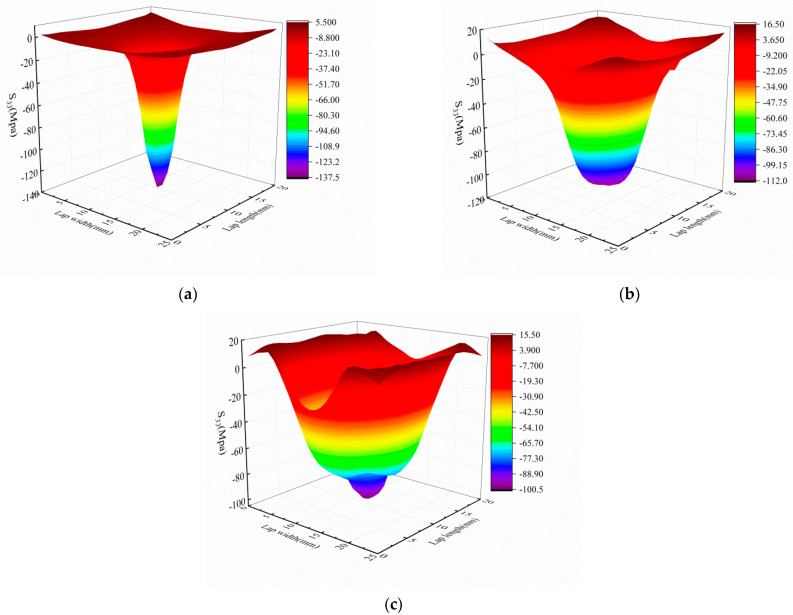
The normal stress distribution of the adhesive layer at different loading points of SLJs under 30 J impact. (**a**) 0.2 *F*; (**b**) 0.6 *F*; and (**c**) *F*.

**Figure 17 materials-14-01016-f017:**
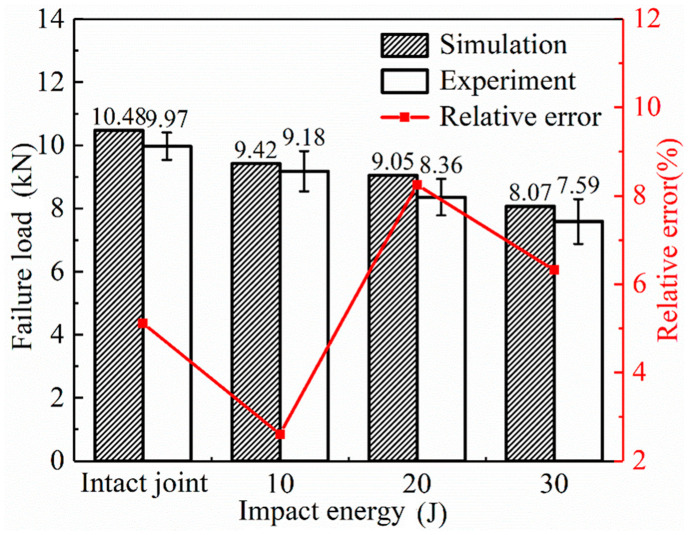
The failure loads of the SLJs under different impact energies.

**Figure 18 materials-14-01016-f018:**
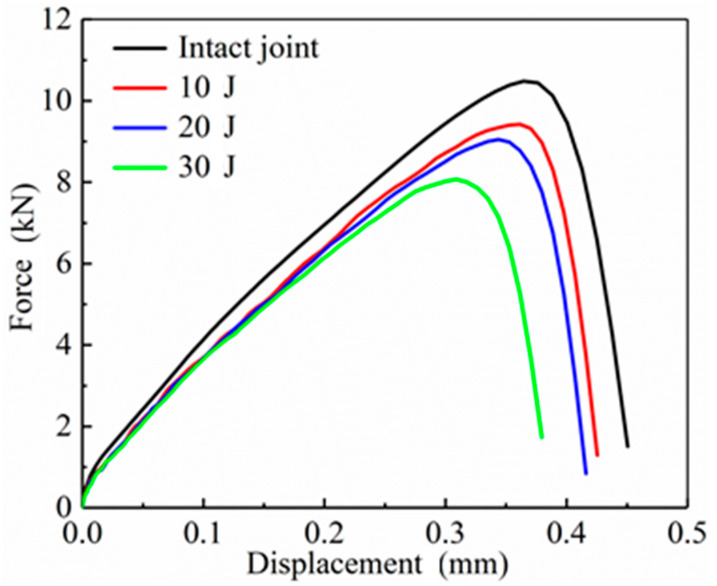
The load–displacement curves of SLJs in axial tension under different impact energies.

**Figure 19 materials-14-01016-f019:**
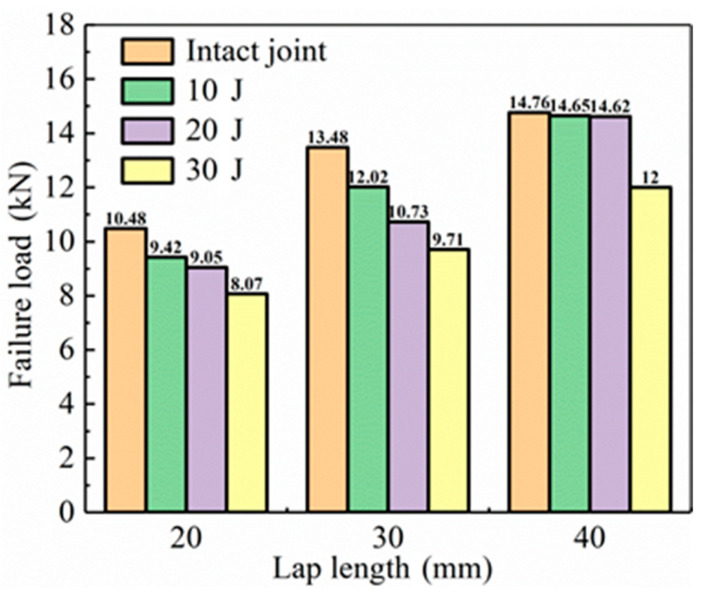
The residual strengths of SLJs with different overlap lengths and impact energies.

**Table 1 materials-14-01016-t001:** Mechanical properties of T300/7901 carbon fiber/epoxy resin composite laminate and adhesive film LJM-170.

Property	CFRP	LJM-170
Young’s modulus *E*_11_/MPa	125,000	2200
Young’s modulus *E*_22_, *E*_33_/MPa	11,300	
Shear modulus *G*_12_, *G*_13_/MPa	5430	815
Shear modulus *G*_23_/MPa	3980	
Poisson’s ratio *v*_12_,*v*_13_	0.3	
Poisson’s ratio *v*_23_	0.42	
Longitudinal tensile strength *X_t_*/MPa	2000	
Longitudinal compressive strength *X_c_*/MPa	1100	
Transverse tensile strength *Y_t_*/MPa	80	
Transverse compressive strength *Y_c_*/MPa	280	
Shear strength *S*/MPa	120	
Interface stiffness *K_nn_*, *K_ss_*, *K_tt_*/(N·mm^−3^)	10^5^	
Maximum normal traction $t_{n}^{0}$/MPa	50	31.9
Maximum shear traction $t_{s}^{0}$, $t_{t}^{0}$/MPa	90	21.2
Toughness in tension $G_{n}^{C}$/(kJ·m^−2^)	0.52	0.48
Toughness in shear $G_{s}^{C}$, $G_{t}^{C}$/(kJ·m^−2^)	0.92	1.83

## Data Availability

No new data were created or analyzed in this study. Data sharing is not applicable to this article.

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
