# Peer review of "Experimental and Numerical Study of Low-Velocity Impact and Tensile after Impact for CFRP Laminates Single-Lap Joints Adhesively Bonded Structure"

_materials, 2021, doi:10.3390/ma14041016_

Round 1

Reviewer 1 Report

The submitted manuscript is dealing with investigation of the influence of low-velocity impact loading on the integrity and the residual life of single-lap adhesively bonded joints. Although the topic is relevant and of high scientific and practical importance, the research goal is unclear and the presented results show numerous deficiencies both in numerical simulations as well as in the experimental part. The convergence of numerical and experimental results is quite poor. The manuscript needs substantial revision before further consideration. The detailed comments are presented below.

1) The failure mechanisms of CFRP structures, especially barely visible impact damage (BVID), under low-velocity impact loading is a key information for understanding and proper analysis of the investigated single-lap joints integrity. Following this, it is recommended to extend the description of the failure mechanisms of BVID in composite structures, in general, and with addressing to the problem of integrity of single-lap joints of composite structures with the support of appropriate references.

2) A part of the literature survey is focused on the influence of elevated temperature on the integrity of single-lap joints. Since the authors do not investigate the influence of temperature, this part seems to be unnecessary and it is proposed to significantly reduce it. On the other hand, the specificity of simulation of single-lap joints using FEM is not well discussed. There are numerous methods and approaches, and numerous papers describing very similar analyses, which are not considered and discussed in the manuscript. Moreover, the issue of integrity and residual strength of single-lap joints after impact loading was not sufficiently presented in the manuscript. It is recommended to extend the literature survey in this direction.

3) The previous comments are directly connected with the weakly formulated research problem. The authors should put their efforts on showing the originality of the performed study with respect to the available literature (which should be discussed earlier according the provided comments above).

4) Please indicate why the described stacking sequence was chosen? Is it connected with a specific application of the investigated CFRP structure?

5) All the variables need to be presented with italic font.

6) It is recommended to change “Value” to “CFRP” in Table 1 for the consistency purpose.

7) Line 106: don’t you meant “mass” instead of “quality”?

8) It is recommended to change in the manuscript the case of 0 J impact energy to “intact specimen” or similar, since, in fact, the specimen from this group was not subjected to any impact loading.

9) The images in Figure 2 need to be enlarged to increase their readability.

10) Please justify the selection of the Hashin criterion among the variety of failure criteria available for the laminated composite structures.

11) Please provide the source for the formulas (7)-(10).

12) Please also justify the selection of CZM for simulation of the contact between laps and their delamination as well as assuming the bilinear constitutive model.

13) Please justify and/or provide the source for the assumed value of parameter $\eta$ in line 235.

14) Line 246: it is suggested to change “0 thickness” to “zero-thickness”.

15) Please clarify which experimental curves are presented in Figure 5: in line 112, the authors mentioned that there were 5 repetitions for each considered impact energy. Which of them are presented on Figure 5? Selected or averaged or other? Since the authors collected such data, it would be interesting to see all 5 repetitions on the plots in Figure 5.

16) It seems that the simulation results disagree, in general, with the experimental results when the impact energy reaches 30 J. This disagreement need to be deeply investigated. Does this happened for all five cases obtained from experiments for 30 J?

17) Figure 8: joining the curves is not appropriate in this case, since the function are expected to be nonlinear. Moreover, the obtained numerical results are highly questionable due to the mentioned disagreements observed for the lap length of 20 mm. In this case, the experimental validation would be necessary to confirm the drawn conclusions on the observed phenomena.

18) The results presented in Figure 9 present surface damage only. This makes them incomparable with the results obtained from the numerical simulations. In order to perform such a comparison one needs to perform NDT testing using one of the methods which makes it possible to show internal damage, e.g. ultrasonic testing, eddy current testing, X-ray computed tomography, etc.

19) The analysis presented in section 4.2.2 has mainly qualitative character, however, to obtain valid conclusions, it is necessary to show the quantitative relationships between impact energy and damage.

20) What is the “non-destructive joint”?

21) The images presented in Figure 13 need to be enlarged, in the current size the described impact damage is not well visible.

22) Figure 15 shows total delamination of the lap joint, which is not true compared to the experiments. Moreover, there is no information about delamination in the tested specimens, again, the non-destructive inspection using the methods mentioned in the comment 18 would be helpful.

23) The conclusions need to be reworked. In the present form they are trivial in some points (i.e. the drawn conclusions are directly follow from the theory of fracture mechanics) and not supported by the experimental results (see the comments above).

24) The language need to be corrected: there are incorrect word endings, lacks of words, grammar and punctuation errors.

Reviewer 2 Report

The paper represents modelling end experimental efforts to capture damage in single lap joints (SLJ) of CFRP laminates. FEM cohesive zone model (CZM) is used to model the damage following the Hashin damage criteria. The model is validated by comparing the dynamic force, system kinetic energy and residual strength of the SLJ. The paper reaches excellent correspondence between model and experiment and given its systematic approach with relatively broad scope (3x load, 3xlap length(one for experiment)) and represents an interesting contribution to the study of SLJ damage. The clear charaterization of the damage as obtained by the model is of a high importance. English needs only minor polishing and paper contains relevant references. Some parts however need clarification to facilitate understanding by the reader from general material science audience, especially the differentiation between experiment and model should be explicitly given.
Major points:
The experimental and FEM data must be clearly identified in all section and figure captions, examples:
4.1.2. – these are model data and should be indicated as such, experimental data may be added to Fig. 8 
Figs10 11 14 15 16are models, Fig18 is experiment

4.2.1. – DAMAGEFT and DAMAGEMT should be related to damage variables introduced in the paper on page 5 

Line444 – SDEG should be related to damage variables on page 5 and 6

Minor points: 

For all device indicate model (make, country)
L41 – surplus -> residual ???
L106 – Quality -> weight or mass ???
L222 - The cohesive unit after suffer damage -> The constiutive relation of damaged cohesive unit (btw. Is there any particular reason to using the term UNITS over the more commonly used ELEMENT?)
L266 – “in this work” should go either to beginning or end of sentence.
L286 most frequent - > most intensive ?
L349 – “the increase decreases gradually”, please rephrase, the question is whether the trend is monotonic as claimed, there seems to be increase from 20 to 30mm and slight decrease thereafter
L370, indicate experiment
L375 - under the 375 illumination of the light.   – illuminated by visible light ?
L385 – indicate modelling
L388 – significant -> higher than ?
L539 – certain -> fixed, given ?

Reviewer 3 Report

The paper entitled "Experimental and Numerical Study of Low-Velocity Impact 1 and Tensile after Impact for CFRP Laminates Single-lap Joints 2 Adhesively Bonded Structure” falls within the scope of the Materials Journal and shows some technical relevance.

The manuscript provides interesting information, and it is based on the experimental study (low-velocity impact and tensile after impact tests), and FEM simulation in order to investigate the mechanical behavior of the single-lap joints adhesively bonded structure of CFRP laminates.

The paper is very well written and presented, includes an adequate amount of data. Furthermore, the organization of idea and flow is, in general terms, appropriate in this manuscript. Although it has technical relevance and main parts of the content could be publishable, it requires certain improvement. In this sense, some suggestions on the attached paper should be addressed before publishing.

Suggestion 01

The authors have done a great job in the literature review. However, the “Introduction” section needs to be improved by including the limitations of the study. It is advisable to introduce a paragraph at the end of that same section clearly describing the objectives of the study where the limitations of the study should also be incorporated.

Suggestion 02

Furthermore, although the authors claim that “the dynamic analysis of the LVI and the residual strength of TAI was not comprehensive and systematic”, the expression is somewhat ambiguous, and novelty is unclear. What is the original contribution of the study? The introduction section is not very enlightening on the subject. Novelty should be made as clear as possible. Please re-organize the introduction part and clear highlight your innovations and contributions. Current version is not enough clear in this issue. Need to show the innovation of your work.

Suggestion 03

The “Materials and Methods” section is well described and would generally allow reproducibility of experiments. However, this section should contain not only the description of the process of each experiment and the model of each tool, but also the standard followed in the test protocol as well as the accuracy of each machine. Please include this information in the revised version of the manuscript.

Suggestion 04

Notwithstanding the readability of the text, considering the FEM study it would be advisable, in order to validate the simulation, clearly define all the boundary conditions and even to confront it against other published data.

This reviewer does not consider that any further modification or clarification is necessary in the remaining sections: "Results and Discussion" and "Conclusion".

Round 2

Reviewer 1 Report

The authors provided detailed answers to the questions from the review report and made corrections and extensions, improving the manuscript. However, several serious issues still need to be clarified or corrected.

1) In the Introduction, the authors mentioned CZM method for modeling damage in composites, but besides this popular approach there is an additional one, equally important – the tiebreak approach. Please provide appropriate description.

2) Figure 8: as it was previously noted, the points should not be connected with lines if the function has not linear character (otherwise, the authors should prove the linearity of these functions). It is obvious that experiments take time and money, if the authors have no possibility to perform the mentioned experiments, they should not draw conclusions on the effectiveness of the numerical models without experimental validation. If the strong argumentation on a possibility of draw such conclusions (e.g. based on literature and/or other experimental studies), all the conclusions about the single-lap joints with the overlap length other than 20 mm need to be removed.

3) Please provide information on the changes made related to the qualitative analysis of the results presented in section 4.2.2.

4) The “no impact joints” is not a correct phrase, I suggest “intact joint” instead.

5) The delamination presented in Figure 15 is not experimentally justified. A conclusion that the other paper has similar results is applicable only if the same joints (material, parameters of joining, overlapping, etc.) were considered. The extended discussion need to be provided in the manuscript regarding the mentioned issues to prove the validity of the results.

Reviewer 3 Report

Since the authors have responded adequately to all comments made on the initial version of the paper, this reviewer has no further objection to its publication in the present form. Nice work.

Author Response

On behalf of all co-authors, I thank you again for reviewing our articles during your busy schedule. At the same time, thank you for your recognition of our work.